# Are Alterations in Skeletal Muscle Mitochondria a Cause or Consequence of Insulin Resistance?

**DOI:** 10.3390/ijms21186948

**Published:** 2020-09-22

**Authors:** Amanda J. Genders, Graham P. Holloway, David J. Bishop

**Affiliations:** 1Institute for Health and Sport (iHeS), Victoria University, Melbourne 8001, Australia; David.Bishop@vu.edu.au; 2Dept. Human Health and Nutritional Sciences, University of Guelph, Guelph, ON N1G 2W1, Canada; ghollowa@uoguelph.ca

**Keywords:** mitochondria, insulin resistance, skeletal muscle, type 2 diabetes, fatty acid oxidation, lipid metabolites, mitochondrial content, mitochondrial function

## Abstract

As a major site of glucose uptake following a meal, skeletal muscle has an important role in whole-body glucose metabolism. Evidence in humans and animal models of insulin resistance and type 2 diabetes suggests that alterations in mitochondrial characteristics accompany the development of skeletal muscle insulin resistance. However, it is unclear whether changes in mitochondrial content, respiratory function, or substrate oxidation are central to the development of insulin resistance or occur in response to insulin resistance. Thus, this review will aim to evaluate the apparent conflicting information placing mitochondria as a key organelle in the development of insulin resistance in skeletal muscle.

## 1. Introduction

In response to increased blood glucose levels, such as after a meal, insulin is released by the β-cells of the pancreas. Insulin then activates a number of different signalling pathways in various tissues [1], with the primary aim of facilitating nutrient uptake and storage in tissues, particularly skeletal muscle, liver, and adipose tissue. Skeletal muscle is responsible for approximately 80% of glucose uptake during a hyperinsulinaemic euglycaemic clamp in healthy individuals [2]. When a normal or elevated insulin level produces an attenuated biological response, this is defined as insulin resistance [3]. In muscle, insulin resistance impairs glucose uptake and, together with impaired suppression of glucose output by the liver, this results in hyperglycaemia and hyperlipidaemia—two factors responsible for many of the co-morbidities associated with insulin resistance and type 2 diabetes (T2D).

Genetic susceptibility and lifestyle factors, such as poor diet, obesity, and physical inactivity, are believed to be risk factors for T2D [4]. Many studies have shown that people with T2D, a family history of T2D, obesity, and/or insulin resistance, have decreased skeletal muscle mitochondrial respiratory function and/or content [5,6,7,8,9]. Humans with insulin resistance and T2D also have a decreased expression of genes encoding key enzymes involved in fatty acid oxidation and the tricarboxylic acid (TCA) cycle, as well as components of the respiratory chain [10]; a lower abundance of mitochondrial proteins has also been reported [11]. In addition, some studies have reported a correlation between insulin resistance and various mitochondrial characteristics [5,7]. However, there is considerable conflict in the literature and whether alterations in skeletal muscle mitochondria are a cause or consequence of insulin resistance is yet to be established [12].

The aim of this review is to present the evidence for and against a central role for mitochondria in the development of insulin resistance. We will also discuss the potential role of altered substrate oxidation by the mitochondria as a contributor to insulin resistance. As exercise training is well established as an effective intervention to improve a range of mitochondrial characteristics [13,14,15], and beneficial in the treatment and prevention of T2D [16,17], we will also examine the link between training-induced changes in mitochondrial characteristics and insulin resistance.

## 2. Mitochondria

Mitochondria are double-membrane organelles present in the majority of cells in the human body. They are responsible for most cellular energy production via oxidative phosphorylation (OXPHOS). In addition, they are involved in many essential cell functions related to homeostasis and cellular metabolism, including intracellular calcium buffering and induction of apoptosis [18,19]. Mitochondria contain their own genome (mitochondrial DNA, mtDNA), which encodes 13 polypeptides of the electron transport chain (ETC), as well as a number of transfer and ribosomal RNAs crucial to mitochondrial function [20]. However, most of the mitochondrial proteome (~1600 proteins) is encoded by the nuclear genome and both genomes are required for functioning mitochondria. Although often represented as oblong independent organelles, skeletal muscle mitochondria form a large interconnected network allowing for the sharing of components and removal of damaged components. The joining and separating of mitochondria from the network is known as fusion and fission, respectively. This is a highly dynamic process that helps mitochondria respond to cellular stresses [21]. In skeletal muscle, mitochondria located just under the sarcolemma are called subsarcolemmal (SS) mitochondria, while mitochondria located between the contractile filaments are referred to as intermyofibrillar (IMF) mitochondria [22] (Figure 1). These two mitochondrial populations are believed to respond differently to contraction and metabolism [23]. For example, while subsarcolemmal mitochondria make up just 10 to 15% of the mitochondrial pool, they are more likely to adapt to variations in muscle use [20]. Thus, the two mitochondrial populations may have unique relationships with insulin resistance [22].

### Mitochondrial Terminology

There is considerable variability in the terminology used to describe mitochondrial characteristics, as well as the techniques used to measure these characteristics. For example, despite its widespread use in the literature, there is currently no widely accepted definition of “mitochondrial biogenesis” [13,24,25] and no consensus on how it is best measured. Given its etymological meaning, we have proposed to define mitochondrial biogenesis as “the making of new components of the mitochondrial reticulum” [25]. It has been suggested that mitochondrial biogenesis can best be assessed by measuring the rate of mitochondrial protein synthesis (mitoPS) [25,26], rather than measurement of changes in the expression of genes and proteins. Nonetheless, while measurement of the synthesis rate of mitochondrial proteins is indicative of mitochondrial biogenesis [25], a more comprehensive assessment of mitochondrial content, structure, quality, and respiratory function, is required to interrogate the relationship between changes in mitochondrial biology and insulin resistance [24,27].

Mitochondrial characteristics include content, size, cristae density, and function. In this review, we will focus predominately on mitochondrial content and respiratory function, as these characteristics have been more extensively investigated. Mitochondrial content refers to the overall volume or density of mitochondria within a muscle or tissue. This can be measured directly using transmission electron microscopy (TEM) or indirectly by measuring the activity of enzymes (e.g., citrate synthase), cardiolipin content, or mtDNA content [28]. TEM can also be used to measure mitochondrial size and cristae density. Mitochondrial respiratory function refers to the ability of mitochondria to use oxygen to generate cellular energy. A search of the literature shows that this can be measured in a variety of ways. This includes direct measures of mitochondrial respiration in isolated mitochondria or permeabilised muscle fibres in vitro; this is commonly done by using either the Seahorse XF Analyzers (Agilent) or Oroboros O2k (Oroboros Instruments). This can yield measures of mass- (per gram of tissue for example) or mitochondrial- (per volume of mitochondria) specific respiration. Of direct relevance to a number of studies cited in this review, respiratory function can also be measured in vivo. A commonly used example is phosphorus magnetic resonance spectroscopy (^31^P MRS), which measures the concentrations of phosphate metabolites and includes measures such as phosphocreatine and ADP recovery time [29,30]. Other techniques used to describe or determine mitochondrial respiratory function include in vitro assays, such as mitochondrial ATP production rate (MAPR); however, there is some concern over whether the findings obtained with these in vitro techniques represent what would occur in vivo [30].

In the literature, mitochondrial changes that are not considered favourable are often described as ‘mitochondrial dysfunction’. However, this term has not been precisely defined and decreases in mitochondrial respiratory function measured in permeabilised or isolated mitochondria in vitro or via ^31^P MRS in vivo, as well as decreases in mitochondrial content, size, or density, or the activity of individual enzymes, have all been used as evidence of ‘mitochondrial dysfunction’ [5,8,31,32,33]. The fact that mitochondria have many functions further highlights the lack of precision provided by the term ‘mitochondrial dysfunction’. Other terms not consistently used between studies to describe similar mitochondrial changes include ‘mitochondrial deficiency’, ‘mitochondrial derangements’, and ‘decreased mitochondrial capacity’. This lack of a consensus, as well as the use of the term ‘mitochondrial dysfunction’ to describe a range of different mitochondrial changes, undoubtedly contributes to confusion regarding the relationship between mitochondrial changes and various indices of health. In this review, we have tried to avoid generic terms, such as ‘mitochondrial dysfunction’. Instead, where possible, we have tried to refer only to the actual measurements made (e.g., changes in mitochondrial content or respiratory function).

## 3. Insulin Resistance

Another area relevant to this review, where the terminology can be unclear, relates to the use of the terms ‘insulin resistance’, ‘insulin sensitivity’, and ‘glucose/insulin (in) tolerance’. These terms, like those associated with mitochondria, have arisen in part due to the use of different methods and models. The gold standard for the measurement of insulin action in vivo is the hyperinsulinaemic euglycaemic clamp. This technique involves the infusion of insulin at a steady rate to maintain hyperinsulinaemia (sometimes a priming dose of insulin is used), which suppresses glucose output by the liver and encourages glucose uptake. Euglycaemia is maintained by a variable glucose infusion, and this rate of glucose infusion indicates whole body insulin action. People or animals requiring greater infusions of glucose are said to be insulin sensitive, while lower rates of glucose infusion are required where insulin action is impaired, e.g., with insulin-resistant participants. When used in conjunction with radioactive tracers, glucose uptake into peripheral tissues, such as skeletal muscle, can also be determined [34]. However, a clamp can be invasive and not practical for some studies. Thus, some studies have used calculations such as the homeostatic measure (HOMA) or the Matsuda index, or similar calculations, which, at least in human populations, correlate well to clamp-based measures of insulin sensitivity or resistance [35]. The use of these indices in rodents is not recommended, as they do not appear to correlate as well [34]. Some studies also use measurements of glycated haemoglobin (HbA1c) to give an estimate of blood glucose concentrations over an extended period. Alternatively, oral and intraperitoneal (i.p.) glucose tolerance tests or i.p. insulin tolerance tests can be used to provide measures of ‘glucose tolerance’ and ‘insulin tolerance’ respectively.

## 4. Mitochondrial Content and Insulin Resistance

A number of studies have found that non-diabetic obese people, and those with T2D, have a decrease in mitochondrial content and size [5,7,36,37,38,39,40]. This has led to the hypothesis that decreased skeletal muscle mitochondrial content is responsible for, or contributes to, the development of insulin resistance [41,42]. If this were true, we would expect to consistently see a lower mitochondrial content in patients with T2D. However, this is not always the case [32,43,44]. It may be that mitochondrial content by itself does not directly influence insulin resistance, but instead may be indirectly involved in some instances; this could be, for example, via alterations in reactive oxygen species (ROS) and/or bioactive lipids. This will be discussed later in the review. In addition, a contributor to the contrasting findings may be the different methods used to assess mitochondrial content and this is discussed first.

### 4.1. Methods of Mitochondrial Content Measurement

The gold standard for the measurement of mitochondrial content is TEM. However, citrate synthase (CS) activity correlates well with TEM-derived measures of mitochondrial content [28], is more readily measured in most laboratories, and is commonly used as an indirect marker of mitochondrial content. Other commonly used measures of mitochondrial content, such as mtDNA content, may be less reliable - at least in healthy individuals [28]. Some studies have instead examined gene expression of mitochondrial proteins, particularly components of the ETC or proteins known to be involved in mitochondrial biogenesis (e.g., PGC-1α). However, although these may give useful information, expression of individual proteins or RNAs does not provide a valid assessment of mitochondrial content [25]. Therefore, the different techniques used, as well as the reliability of these measures, may explain some of the variability in findings as well as biological variation observed within studied populations.

### 4.2. Mitochondrial Content in Patients With Insulin Resistance or Type 2 Diabetes

As alluded to earlier, the evidence indicating that skeletal muscle mitochondrial content is lower in patients with insulin resistance and/or T2D is not conclusive; although many studies have shown lower mitochondrial content, some have not (Table 1). Using CS activity as an indirect marker, a lower mitochondrial content was detected in T2D, but not obese men [5], although this value was normalised to creatine kinase activity and therefore difficult to compare to other studies. However, there may be sex-specific differences as CS activity was observed to be lower in obese women (insulin resistance not reported) when compared with lean women [37]. An alternative indirect marker of mitochondrial content, mtDNA content, wasn’t different between diabetic and nondiabetic subjects in one study [32] but was lower in T2D subjects compared with controls in another study [38]. It has also been reported that in contrast to long-term diagnosed T2D patients, mitochondrial content was not significantly different in insulin-resistant (not T2D) participants when compared to BMI and age-matched controls [39]. However, a study in Asian Indian participants found that despite displaying a high level of insulin resistance these participants also had increased mitochondrial content [45]. Thus, although many studies have reported a lower mitochondrial content and/or size, this finding is not consistent even when using the same method (e.g., TEM). From this data, it is also difficult to determine whether a decrease in mitochondrial content precedes or follows the development of T2D in humans.

In addition to changes in overall mitochondrial content, alterations in mitochondrial distribution may occur with type 2 diabetes. Thus, TEM has also been used to examine changes to specific mitochondrial populations, but again without consistent results. In one study in non-diabetic, insulin-resistant participants and T2D patients, both groups had lower IMF, but not SS, mitochondria content relative to insulin-sensitive lean participants (~20% lower in insulin-resistant participants, and ~40% lower in T2D patients) [36]. Another study reported lower SS mitochondrial content in T2D patients in comparison to lean controls [7]. Yet, another study found no difference between the content of SS and IMF mitochondria in T2D participants compared to controls [46]. Studies examining mitochondrial size via TEM have also found either no change [36] or a decrease in mitochondrial size in both obese and T2D participants [5]. Thus, it is unclear from studies in human participants whether changes in mitochondrial size or distribution occur with insulin resistance. Other studies have examined this relationship in animal models and this will be discussed later.

Studies in participants with a family history of T2D have attempted to determine whether a decrease in mitochondrial content precedes T2D. In young men and women with a family history of T2D, mitochondrial content (as measured by TEM) was decreased compared to controls [6]. In young men with a family history of T2D, skeletal muscle mtDNA content was decreased in comparison to men without a family history [47], suggesting the mitochondrial content changes occur prior to the development of T2D. The results from these two studies indicate that a decrease in mitochondrial content may precede the development of T2D. However, there could be other explanations for a decreased mitochondrial content, including increased physical inactivity and obesity, as discussed below.

### 4.3. Correlations between Mitochondrial Content and Insulin Resistance

The potential link between lower mitochondrial content and insulin resistance has also been examined by assessing the correlation between these two measures. Lower intermyofibrillar, but not subsarcolemmal, mitochondrial content has been reported to be strongly correlated with glucose disposal during a euglycaemic clamp in a group of participants that included individuals who were lean insulin-sensitive, insulin-resistant, and or who had been diagnosed with type 2 diabetes [36]. Lower mitochondrial size has also been correlated with greater insulin resistance [5]. Furthermore, increases in skeletal muscle mitochondrial content (measured by cardiolipin content and CS activity) have been reported to accompany improvements in hyperglycaemia in T2D patients [48]. However, Phielix et al. [9] found no relationship between mitochondrial content and insulin sensitivity; in this study, mitochondrial content was measured by mtDNA content, which as noted above, has been criticised as valid method to estimate mitochondrial content. Thus, studies in humans suggest that mitochondrial content is correlated with insulin resistance, although this does not provide evidence of a causative link between the two variables.

### 4.4. Interventions That Alter Mitochondrial Content and Their Effects on Insulin Resistance

Skeletal muscle mitochondrial content is known to be strongly influenced by physical activity, with an increase in content with exercise training in both healthy individuals [13,49] and those with insulin resistance [8,9,50,51,52,53,54]. In contrast, a decrease in mitochondrial content occurs following manipulations that decrease physical activity, such as bed rest [55,56]. Thus, some researchers have manipulated physical activity and investigated whether changes in mitochondrial content and insulin resistance occur in parallel (Table 2).

In obese, but not diabetic, participants exercise training increases mitochondrial content as assessed by increases in mtDNA content [9], the activity of ETC enzymes [52], CS activity [48,57,58], and TEM [59,60], alongside improving measures of insulin resistance and glucose tolerance. In T2D patients, exercise training can similarly improve insulin resistance and decrease hyperglycaemia [61] while increasing mitochondrial content as measured by TEM [46] or surrogate measures (e.g., mtDNA, CS activity) [9,61,62]. This would indicate exercise training increases mitochondrial content alongside whole-body improvements in insulin resistance. In contrast, a study in sedentary T2D patients found that although some measures of mitochondrial content were increased post-training, there was no change in blood glucose concentration [50]. However, the lack of improvement in blood glucose in that study may have been due to insufficient exercise frequency [16,17]. Exercise-induced improvements in insulin resistance were recently shown to disappear after four non-exercise days [63], while other studies have reported a much slower decline in mitochondrial content [49]. This suggests that although exercise training is able to improve insulin resistance and increase muscle mitochondrial content, exercise-induced improvements in insulin resistance may be related to acute effects rather than longer-term changes such as alterations in mitochondrial content. This would support the hypothesis that there isn’t a direct relationship between mitochondrial content and insulin resistance.

Conversely, weight loss, which is able to improve insulin sensitivity in insulin-resistant patients, actually decreases mitochondrial size without changing mitochondrial content (measured by TEM) [60]. This led the authors to conclude that improvement of insulin sensitivity is not dependent on an increase in mitochondrial content, but that changes in mitochondrial content are more likely linked to altered physical activity levels [60]. Thus, in humans, skeletal muscle mitochondrial content does appear to change alongside changes in insulin resistance (Figure 2). However, there is limited evidence for a direct relationship between the two variables.

### 4.5. Experimental Animal Models

As studies in human participants have not been able to confirm whether decreases in skeletal muscle mitochondrial content cause insulin resistance, animal models have been used to investigate the relationship between changes in insulin resistance and mitochondrial content. Animal studies can more tightly control factors that can influence mitochondrial content such as physical activity, diet, and body mass. Animal studies can also be used to manipulate mitochondrial content. For example, overexpression of PGC-1α induces mitochondrial biogenesis and improves insulin-stimulated glucose transport in lean and obese Zucker rats [70]. A recent study in Goto-Kakizaki (GK) rats examined various mitochondrial characteristics in comparison with control rats in response to hind-limb contraction. GK rats have impaired β-cell insulin secretion and develop T2D at approximately three weeks of age. This is followed by peripheral insulin resistance and hepatic glucose overproduction. GK rats are not obese and are physically active. Despite developing T2D, this study found that there was no change in mitochondrial content (measured by cytochrome c content) between the control and GK rats [71]. Zucker diabetic fatty (ZDF) rats, another model used to study insulin resistance, have increased number, width, and content of subsarcolemmal mitochondrial compared to controls; however, intermyofibrillar mitochondria are unchanged and CS activity is not different to controls [72]. In red muscle of Zucker rats, intermyofibrillar mitochondria are larger in the obese animals compared to lean animals; however, this size difference is not observed in the subsarcolemmal mitochondria nor in either mitochondrial population in the white muscle. Mitochondrial content was increased in the red muscle of the obese Zucker rats compared to lean animals [73]. Exercise training can further increase mitochondrial content in these animals alongside improving hyperglycaemia, although insulin-mediated glucose uptake is not improved [74]. Therefore, genetic animal models demonstrate that insulin resistance can occur in the absence of changes in mitochondrial content.

Dietary models of insulin resistance (over nutrition or consumption of diets high in sucrose or fat) are often used in rodent studies as they can more closely mirror the development of disease in humans. In Wistar rats fed a high sugar diet, there are alterations of the mitochondrial ultrastructure as well as an increase in CS activity [75]. Regular physical activity in these rats partially returned mitochondrial ultrastructure towards the control values, although CS activity was not significantly altered with exercise in the animals consuming a high sugar diet [75]. This suggests improvements in mitochondrial content were not responsible for the improvements in glucose tolerance seen with exercise training in these rats [75]. The feeding of a high fat diet (HFD) is a commonly used model of the metabolic syndrome in rodents, as they develop insulin resistance, compromised β-cell function, and greater total body and fat mass [76]. In rodents fed a HFD, contrasting findings have been reported for markers of mitochondrial content, including mtDNA content, expression of OXPHOS proteins, and CS activity, with some studies showing increased mitochondrial content [77,78,79,80], no change [81], or a decrease in mitochondrial content [82]. These findings provide further evidence a decrease in mitochondrial content per se is not responsible for insulin resistance, at least in rodent dietary models of insulin resistance.

### 4.6. Conclusions

A lower skeletal muscle mitochondrial content has typically been observed in people with insulin resistance and type 2 diabetes. However, there is not sufficient evidence from these studies, nor those in animal models, to suggest this reduction in mitochondrial content has a causative role in the development of insulin resistance. It has also been argued that the decrease seen is not sufficient to cause insulin resistance, as skeletal muscle has enough mitochondria to allow for a large and sufficient increase in substrate oxidation and ATP output during exercise [83,84]. Thus, lower skeletal muscle mitochondrial content is not likely to be directly responsible for the development of insulin resistance and T2D. Rather decreased mitochondrial content may occur in conjunction with other changes that contribute to insulin resistance, such as an increase in bioactive lipids and ROS [85]. It has also been proposed that some of the conflicting findings between animal models and humans may be associated with the stage of diabetes development and circulating insulin concentrations [86]. However, although total mitochondrial content is important, equally important is how well these mitochondria work; i.e., their respiratory function.

## 5. Mitochondrial Respiratory Function

Mitochondria have many “functions”. In this section, we will focus on one of the most important functions of mitochondria—to use oxygen to generate cellular energy. This is often measured via maximum mitochondrial respiration (oximetry), but, as mentioned in the ‘mitochondrial terminology’ section, this has also been measured using other techniques - some more representative than others. As noted by Lewis et al. [87] and Holloszy [83], the many ways mitochondrial respiratory function has been measured in the literature has likely contributed to some of the conflicting findings. For example, use of different substrates or physiological or saturating ADP levels, or resting observations when mitochondrial respiration is controlled by ATP demand [83,88], may have contributed to some of the inconsistent findings.

A number of arguments have been made to counter the proposed central role of lower mitochondrial respiratory function in the development of skeletal muscle insulin resistance. These arguments focus on a number of factors, such as the large spare respiratory capacity in skeletal muscle (reviewed by Holloszy [83,89]), the influence of the method of mitochondrial respiratory function measurement (reviewed in Lewis et al. [87]), and studies in disease and knockout models in which mitochondrial respiratory function is decreased but insulin resistance is absent [90,91]. In addition to measurement differences, it has been suggested that mitochondrial respiratory function is not related to insulin sensitivity, and that the joint observation of changes in mitochondrial content and/or respiration and insulin resistance is coincidental or linked by a common factor (s), which might include physical inactivity, obesity, or reduced skeletal muscle blood flow [92,93]. It has been further suggested that a decrease in mitochondrial respiration is a consequence rather than a primary cause of the altered cellular metabolism and insulin resistance that develops with nutritional overload [33,94,95].

### 5.1. Mitochondrial Respiratory Function in Patients with Insulin Resistance or Type 2 Diabetes

Previous studies have reported that insulin-resistant or T2D patient populations have lower mitochondrial respiratory function [5,7,42,43,92,96,97,98,99]. For example, people with T2D have lower mitochondrial respiratory function than controls, as measured by both ^31^P MRS [8] and high-resolution respirometry in permeabilised muscle fibres [9]. However, other studies, using a variety of measurement techniques, have either not found reductions in mitochondrial respiratory function or found that observed decreases could be attributed to decreases in mitochondrial content [32,38,100,101]. Thus, although studies have observed that patients with T2D or even insulin resistance display lower mitochondrial respiration, there are also studies with contradictory findings. Thus, it is not clear whether mitochondrial respiration, independent of other contributing factors, particularly physical activity, is lower in T2D or indeed responsible for the development of insulin resistance.

A potential cofounder contributing to the dissociation between mitochondrial function and insulin resistance is the duration of T2D when the measurements are made [88]. One study observed a decrease in mitochondrial respiration only in sedentary long-standing T2D patients, which suggests that mitochondrial respiration does not precede or cause insulin resistance [39]. Several studies have also examined the non-diabetic offspring of T2D parents. Petersen et al. [97] found that mitochondrial respiratory function, as measured by ^31^P MRS, was decreased in insulin-resistant offspring [97], although they did not control for mitochondrial content. However, skeletal muscle MAPR assessed during a luciferase-based assay was not different between children of T2D mothers and controls [66]. Therefore, evidence of a decrease in mitochondrial function being an inherited defect leading to the development of insulin resistance and T2D is inconsistent and requires further research.

### 5.2. Is There a Relationship between Mitochondrial Respiratory Function and Insulin Resistance?

Some studies have attempted to correlate mitochondrial respiration to insulin resistance. In a study of control and T2D participants, no correlation was found between markers of insulin sensitivity or insulin-stimulated glucose disposal and mitochondrial respiration [9]. In contrast, in non-diabetic but insulin-resistant participants, mitochondrial respiration was correlated with insulin resistance. However, after adjusting for physical activity and/or trunk fat mass, this relationship between muscle insulin sensitivity and mitochondrial respiration was partially lost and no longer significant [98]. In a study examining bed rest (an extreme form of physical inactivity), there was a positive correlation between mitochondrial respiration and insulin sensitivity [55]. However, on further examination of these findings, this effect may have been due to changes in mitochondrial content rather than changes in respiration [55]. Therefore, although some studies have observed parallel changes in mitochondrial respiration and insulin resistance, the relationship between the two appears to be affected by other factors, such as mitochondrial content, physical activity level, and fat mass.

### 5.3. Interventions That Alter Mitochondrial Respiration and Their Effects on Insulin Resistance

#### 5.3.1. Physical Activity

Many T2D patients have a sedentary lifestyle. This has a negative impact on the physical capacity of T2D patients, which is strongly associated with their disease status [102]. It has been demonstrated in many studies of healthy individuals and those with insulin resistance and/or T2D that exercise training can increase mitochondrial respiration [8,9,13,14,15,103] (Figure 3). Exercise interventions that increase mitochondrial respiratory function have also been shown to improve insulin resistance [8,9]. Conversely, bed rest causes insulin resistance as well as reducing maximal and submaximal mitochondrial respiration [56]. Although, when mitochondrial respiration was normalised to CS activity these decreases were no longer statistically different [56]. Despite this, single leg immobilization decreases respiration before any changes in the expression of components of the respiratory chain [104]. This suggests that changes in mitochondrial respiratory function may occur prior to changes in content, which may take a longer time to occur. However, it is currently unclear whether improvements in mitochondrial respiratory function are responsible for the concurrent improvements in insulin resistance or if this occurs via different pathways. Further studies are required to fully answer this question.

#### 5.3.2. Weight Loss

Weight loss plays an important role in the treatment of insulin resistance and type 2 diabetes [17]. Thus, studies have attempted to determine if significant weight loss in insulin-resistant and T2D patients is accompanied by improvements in mitochondrial respiration. In obese women placed on a very low-calorie diet, glucose metabolism was improved but mitochondrial respiration (measured in permeabilised muscle fibres) was decreased [105]. In patients who had gastric bypass surgery (which results in significant weight loss and improvements in insulin sensitivity), mitochondrial respiration measured in permeabilised muscle using high-resolution respirometry was only improved in the patients who also exercised [68]. This suggests a disconnect between improvements in insulin sensitivity and skeletal muscle mitochondrial respiration, at least following weight loss.

### 5.4. Experimental Animal Models of Insulin Resistance and Mitochondrial Respiration

Factors such as physical activity levels and degree of obesity complicate investigations into the relationship between mitochondrial respiratory function and insulin resistance. Some researchers have therefore examined the relationship between mitochondrial respiration and insulin resistance in experimental animal models that can more tightly control factors such as physical activity and body mass.

Many studies in animal models with impaired mitochondrial respiratory function have found that both basal and insulin-stimulated glucose transport are normal or increased [83,90,106]. For example, a strain of skeletal-muscle-specific knockout mice with decreased ETC function have normal insulin tolerance and increased glucose uptake via AMP kinase (AMPK) activation [90]. Reduced activity of complex I of the ETC and decreased state-3 respiration in isolated mitochondria in skeletal muscle homogenates has been reported following treatment with metformin and thiazolidinediones, well known for improving insulin sensitivity [107]. In addition, there is increased mitochondrial respiratory function in skeletal muscle from rats with streptozotocin-induced hyperglycaemia [106]. However, Petersen and Shulman [12] have suggested this could be explained by the upregulation of other pathways (such as glycolysis or increased fat oxidation) to increase or maintain glucose uptake [12]. Overexpression of heat shock protein (HSP) 72 in mouse skeletal muscle increases mitochondrial oxidative capacity but decreases insulin resistance [108]. Thus, while some genetic studies suggest a link between mitochondrial respiration and insulin resistance, the majority do not.

Dietary models have also been used to investigate the link between insulin resistance and mitochondrial respiration. Rats and mice fed a HFD often have increased mitochondrial respiratory function, despite showing insulin resistance and a reduction in glucose tolerance [77,78,79]. Another study found no effects of a HFD on mitochondrial respiratory function when classical complex I and II substrates were used [81]. However, the animals in this study only received the HFD for two weeks, which may not have been sufficient to affect mitochondrial respiration. Measures of insulin resistance were not undertaken. There are, however, studies using a HFD to induce insulin resistance in mice that have reported a decrease in resting and insulin-stimulated soleus muscle mitochondrial respiration [82] and decreased mitochondrial respiration and ADP sensitivity across a range of biologically relevant ADP concentrations [31]. Despite this, rodents fed a high-fat, high-sucrose diet for one month developed glucose intolerance (and diabetes after 16 weeks on this diet) but had no change in mitochondrial respiration [33]. This suggests that decreases in mitochondrial respiration do not precede the onset of diet-induced insulin resistance in mice. While there are some conflicting findings in HFD rodents, these may be related to experimental factors such as differences in diet composition, muscle studied, and method of mitochondrial respiration measurement (e.g., isolated mitochondrial vs permeabilised muscle fibres). What is clear, however, is that diet-induced insulin resistance is not always accompanied by a decrease in mitochondrial respiration and has often been associated with an increase in mitochondrial respiration.

### 5.5. Summary

In conclusion, many studies in humans have observed that mitochondrial respiratory function is lower when insulin resistance is present. However, conflicting data from human participants, and especially animal models, suggest that changes in insulin resistance can occur without corresponding alterations in mitochondrial respiration and vice versa. Thus, there is not sufficient evidence to conclude that a decrease in mitochondrial respiration is a direct cause of insulin resistance. It is more likely that other another factor(s) link insulin resistance and mitochondrial respiration. For example, it has been proposed that changes in mitochondrial β-oxidation lead to increases in intracellular fatty acid metabolites, which disrupt insulin signalling [42]. Thus, in the next section we will discuss the potential role of altered fatty acid oxidation and lipogenesis in insulin resistance. An increase in ROS emission from the mitochondria has also been reported to contribute to insulin resistance. However, discussion of this is beyond the scope of this review, and thus the reader is directed to other relevant reviews for more information on this topic [109,110].

## 6. Fatty Acid Metabolism and Insulin Resistance

Intramyocellular lipid (IMCL) content is increased in people with obesity and T2D [42,92,95]. This could be due to increased fatty acid uptake by the muscle and/or decreased fatty acid oxidation (FAO) in the mitochondria [37,42,111,112,113]. A mismatch between fatty acid uptake and oxidation can also lead to the formation of a variety of fat-derived, potentially toxic, lipid metabolites, such as ceramides, sphingomyelins, acylcarnitines, and diacylglycerols (DAG). These fatty acid metabolites have been hypothesised to be involved in the development of insulin resistance, via impairment of insulin signalling [12,113] and reduced insulin-stimulated glucose uptake by skeletal muscle [112,114,115]. While highly trained endurance athletes also have increased IMCL stores, they have normal insulin sensitivity. This has been attributed to their increased ability to metabolise fat, which has been correlated to their increased skeletal muscle mitochondrial content compared to controls [116]. This aligns with the observation that exercise can prevent fatty acid induced insulin resistance through an increase in fatty acid oxidation, as well as increased triglyceride (TG) synthesis and a reduction in bioactive fatty acid metabolites such as DAG and ceramide species [117]. Thus, a mismatch between fatty acid supply and oxidation by the mitochondria that leads to an increase in lipid metabolites may also be an important determinant for skeletal muscle insulin resistance [12,42,113].

### 6.1. Increased Inter- and Intramyocellular Lipids in Patients with Obesity and Type 2 Diabetes

As noted above, patients with obesity, insulin resistance, or T2D have frequently been shown to have greater IMCL stores. However, highly trained endurance athletes also have higher IMCL compared to less active controls, but normal insulin sensitivity - a phenomenon referred to as the “athlete’s paradox”. This has been explained by the ability of exercise training to improve lipid turnover and lipid droplet (LD) quality and characteristics [118,119], and to increase FAO in the mitochondria [8,50,53,120,121]. Endurance exercise training also promotes high rates of TG synthesis, which alters intramuscular lipid portioning, localisation (by suppling FFA at the site of energy demand), and lipotoxicity [119,122]. Consequently, it appears that endurance exercise training and HFD differentially affect the quantity and composition of DAG molecular species in rat skeletal muscle [123]. Thus, a simple increase in IMCL content cannot be responsible for skeletal muscle insulin resistance; however, it may indicate a mismatch between lipid supply and oxidation.

While the greater IMCL stores per se do not appear to be an important determinant of insulin resistance, the location of lipids within skeletal muscle may be important. Patients with T2D have increased subsarcolemmal lipids, whilst lipids in the intermyofibrillar region are not different to either obese controls or endurance-trained athletes. There is a strong inverse relationship between subsarcolemmal lipid volume and insulin sensitivity, which is improved with exercise training [46]. The subcellular localisation of lipid species also appears to be important, with mitochondrial lipids having the ability to alter mitochondrial respiratory function and ROS production [94,124,125,126]. Perreault et al. [126] investigated the subcellular localisation of DAGs, ceramides, and sphingolipids in the skeletal muscle of endurance athletes, lean participants, obese participants, and participants with T2D. The mitochondrial/endoplasmic reticulum fraction had greater DAG species in lean participants compared to T2D and in athletes compared to obese and T2D, and these were positively related to insulin sensitivity. Within the same fraction, C18:0 ceramide content was related to insulin resistance. However, when cytosolic lipids were examined there was no significant differences between groups and no relationships between cytosolic lipids and insulin resistance. This gives an indication that while overall lipids levels may not change, lipids specifically in the mitochondria [126], and the subcellular localisation of lipids, which is not commonly measured, may be important factors when considering the metabolic impact of increased skeletal muscle lipid content on insulin resistance.

### 6.2. Evidence for a Role of Impaired Mitochondrial Fatty Acid Metabolism in the Development of Insulin Resistance

Studies in obese and/or T2D patients have often found defects in FAO as well as lower expression and activity of enzymes important for FAO. For example, in obese women, palmitate oxidation and β-HAD activity are lower compared with lean women [37]. In a study comparing sedentary, obese, insulin-sensitive, and insulin-resistant women genes involved in lipid droplet and fatty acid metabolism had lower expression in insulin-resistant compared with insulin-sensitive women. For example, gene expression of carnitine palmitoyltransferase 1B (CPT1B), malonyl CoA decarboxylase, hormone sensitive lipase, and adipose triglyceride lipase were lower in insulin-resistant women compared to insulin-sensitive women [127]. The authors also reported that low oxidative capacity of skeletal muscle was associated with ceramide accumulation and insulin resistance [127]. In contrast, studies using isolated mitochondria from participants with obesity and T2D do not show reductions in FAO rate [40,62]. However, it was noted in one of these studies that FAO was reduced at the whole-muscle level [40]. This suggested that either reductions in mitochondrial content were responsible for the lower FAO rate rather than intrinsic reductions in the ability of the mitochondria to metabolise fats, or that only a small proportion of the mitochondrial pool was studied due to the mitochondrial isolation technique used [40]. However, other studies have not found defects in mitochondrial content, respiratory function, or ATP synthesis with acute periods of increased fatty acids [128,129]. Therefore, although there is some evidence FAO is decreased in those with insulin resistance, these findings aren’t uniform and the influence of reduced FAO on the development of insulin resistance is unclear. The connection between reduced FAO and insulin resistance may involve multiple different pathways, including the possibility that lipid-induced insulin resistance results from mechanisms that decrease glucose uptake but which are not linked to an impairment in mitochondrial FAO.

In addition to lower rates of FAO, greater rates of incomplete FAO may also accompany insulin resistance. High lipid availability can lead to high lipid transport into the mitochondrial matrix. However, without greater metabolic ATP demand this can result in incomplete FAO that leads to the accumulation of bioactive lipids such acylcarnitines. For example, it has been found that patients with T2D have increased β-HAD activity but no increase in CS activity. This could lead to a situation whereby there is increased production of acylcarnitines [130]. Higher levels of these bioactive lipid metabolites may then interfere with insulin signalling, glucose uptake, and potentially mitochondrial respiration [80,95,117,131,132]. It has been reported that patients with poorly controlled T2D have greater incomplete FAO in comparison to those whose diabetes is well controlled, and this is associated with higher insulin resistance and higher HbA1c. Furthermore, linear regression found that incomplete FAO explained 40% of the variance in insulin resistance and 69% of the variance in HbA1c levels [130]. Thus, it has been hypothesised that incomplete FAO may be a factor contributing to insulin resistance [12].

Lending support to this hypothesis is that some studies have found that when mitochondrial fatty acid entry is limited, such as by deleting or inhibiting CPT1b in skeletal muscle, there is a corresponding improvement in glucose homeostasis and lipid-induced insulin resistance [95,133,134]. Conversely, a separate study in rats overexpressing CPT1b found insulin action is increased and is sufficient to prevent lipid-induced insulin resistance in skeletal muscle. This was accompanied by a decrease in TG content, and the membrane to cytosolic ratio of DAG. However, markers of mitochondrial content or function were not altered by overexpression of CPT1b nor were there any changes to the skeletal muscle acylcarnitine profiles irrespective of diet [135]. In addition, obese women have decreased CPT1 activity in comparison to lean controls [37]. The reasons for these seemingly conflicting findings are unclear, but it has previously been suggested that this could be due to methodological differences such as length of the HFD, muscles examined or manipulated, and/or method of manipulating CPT1b levels [135]. However, despite these conflicting findings, collectively the literature does suggest that changes in the entry of fatty acids into the mitochondria and changes to FAO may influence glucose homeostasis.

### 6.3. The Influence of Exercise Training and Weight Loss on Imcl Stores and Fatty Acid Oxidation

Exercise training is a common manipulation that is known to improve FAO [117,136,137], to decrease DAG and ceramide accumulation [53,138], and to alter lipid localisation in skeletal muscle [126] alongside improvements in insulin resistance (also discussed above). In contrast, some studies have not found a direct link between exercise-training-induced improvements in muscle lipids and/or fatty acid oxidation and insulin resistance [62,139]. As dietary interventions and significant weight loss are also known to improve insulin resistance, the impact of significant weight loss on IMCL and FAO has been examined both together and independently of exercise training. A study comparing the effects of weight loss and exercise separately in sedentary overweight or obese men found that while insulin resistance, DAG content, body mass, and fat mass were improved by both interventions, muscle TG and lipogenic enzyme content decreased with diet-induced weight loss whilst increasing with exercise. Ceramide and sphingosine content only decreased with exercise training, and not with weight loss only. However, changes in total DAG, ceramides, and sphingolipids did not correlate with changes in glucose disposal [138]. A comparison between patients post gastric bypass surgery found those who also undertook moderate-intensity exercise training saw greater decreases in skeletal muscle ceramides and certain species of sphingolipids and a greater improvement in insulin sensitivity, while TG were decreased to a similar extent [68]. Thus, the results of studies using exercise and diet manipulations that alter IMCL stores, FAO, and lipid metabolites are yet to establish a clear relationship between these factors and insulin resistance.

### 6.4. Evidence from Animal Models

A number of animal models have also been utilised to investigate the potential link between FAO and insulin resistance. While HFD rodents develop insulin resistance, the skeletal muscle from HFD mice has been reported to have an increased capacity for FAO, with either no difference in the ratio of complete to incomplete FAO [79] or increased rates of incomplete FAO and greater accumulation of β-oxidation intermediates [22]. HFD mice also have increased oxidative enzyme activity (β-HAD—β-hydroxyacyl CoA dehydrogenase, MCAD—medium-chain acyl-CoA dehydrogenase, and CS) and protein expression of PGC-1α, uncoupling protein (UCP)3, and mitochondrial respiratory chain subunits, as well as the expression of enzymes and transporters involved in β-oxidation [79,81]. CPT1b, in particular, is increased in the skeletal muscles of rodents fed a HFD [79,81]; this is most likely due to an increase in mitochondrial biogenesis in these animals as CPT1b activity is not changed in isolated mitochondria [80]. This does not appear to be specific to rodents, as consumption of a western diet (containing increased sucrose and fructose, as well as fats) by female cynomolgus macaques increases FAO in comparison to macaques eating a Mediterranean diet [140]. Thus, results from various high fat diet animal models indicate that insulin resistance can occur despite an increase in mitochondrial FAO.

Consistent with the above, previous trials to increase β-oxidation via pharmaceutical or genetic means in mice fed a HFD have been insufficient to protect them from insulin resistance [94,141,142]. In addition, other rodent models of diabetes show increases in FAO, but this does not seem sufficient to protect them from diabetes. For example, rats treated with streptozotocin show increased lipid transport/sensitivity alongside increased glucose concentrations, despite increased fat oxidation [143]. Leptin deficient *ob/ob* mice (a rodent model of type 2 diabetes) also have an increase in mitochondrial proteins associated with β-oxidation compared to lean controls [144]. This suggests that these mice have an increased ability for FAO in skeletal muscle compared to lean controls [144]. Together these results indicate that increases in β-oxidation without a corresponding increase in oxidative phosphorylation as a result of increased ATP demand are not sufficient to protect against insulin resistance. Consistent with human studies, the results from these animal studies have not established a strong link between decreased FAO in the mitochondria and the development of insulin resistance.

### 6.5. Summary

A mismatch between fatty acid uptake and oxidation, which leads to an increase in IMCL content and lipid metabolites, has been observed in individuals with insulin resistance and T2D [42,92,95]. However, the elevated IMCL content in highly trained endurance athletes, a phenomenon referred to as the “athlete’s paradox”, argues against the hypothesis that an increase in IMCL per se is an important determinant of skeletal muscle insulin resistance. However, the subcellular localisation may be an important factor when considering the metabolic impact of increased skeletal muscle lipid content on insulin resistance. Even if total IMCL content is not a direct cause of insulin resistance, it may be an indicator of a potential mismatch between fatty acid supply and oxidation, which, if also associated with an increase in lipid metabolites, may contribute to the development of insulin resistance. The accumulation of lipid metabolites may also explain the observation of reduced fatty acid oxidation, or greater rates of incomplete fatty acid oxidation, in the mitochondria with insulin resistance. However, there are many conflicting findings and more research is required to establish the contribution of changes in fatty acid uptake, fatty acid oxidation in the mitochondria, IMCL content, and lipid metabolites in the development of insulin resistance.

## 7. Conclusions

There is substantial evidence that people who have type 2 diabetes or insulin resistance have reduced skeletal muscle mitochondrial content and respiratory function. Furthermore, both insulin resistance and mitochondrial characteristics can be improved by exercise training. However, despite this seemingly correlative relationship there is insufficient evidence to suggest that mitochondrial content and respiratory function directly affect insulin sensitivity. Instead, there is stronger evidence that insulin resistance may be due to the increase in lipid metabolites that occurs when an increase in the supply of fatty acids is not matched by a commensurate increase in IMCL synthesis and fatty acid oxidation in the mitochondria (which has been associated with mitochondrial content). Future studies investigating the influence of manipulating mitochondrial content, mitochondrial respiratory function, IMCL content, lipid metabolites, and fatty acid oxidation on insulin resistance may prove useful to untangle to complex relationship between mitochondria and insulin resistance.

## Figures and Tables

**Figure 1 ijms-21-06948-f001:**
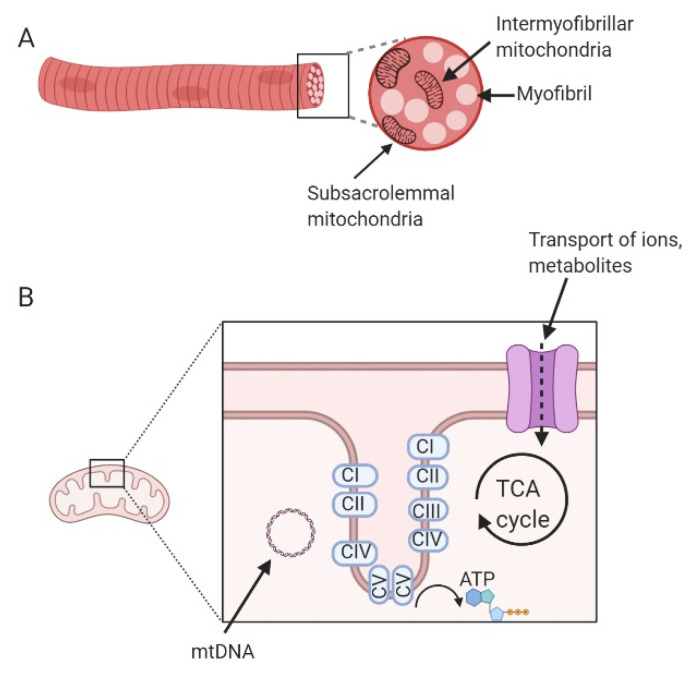
(**A**) Location of mitochondria within the skeletal muscle fibre. (**B**) Location of metabolic processes within the mitochondria. Black boxes indicate a section of each figure that has been magnified in the right-hand image. Created with BioRender.com.

**Figure 2 ijms-21-06948-f002:**
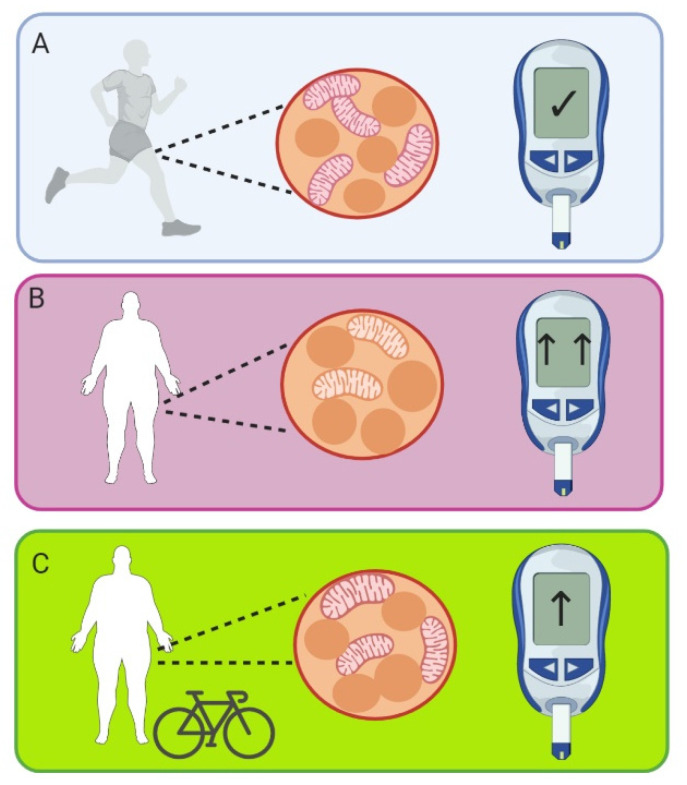
Mitochondrial content in (**A**) fit, healthy people; (**B**) Insulin-resistant people; (**C**) Insulin-resistant people with exercise training. Mitochondrial content tends to correlate with insulin resistance, with good insulin sensitivity in fit, healthy people, and lower mitochondrial content in insulin-resistant people. Both mitochondrial content and insulin sensitivity is improved with exercise training. Glucometer represents the degree of insulin sensitivity, and associated changes in blood glucose concentration. In panel A the glucometer shows blood glucose in the normal range, and fit, healthy people have normal insulin sensitivity. In panel B the glucometer shows the increased blood glucose concentrations present in people with insulin resistance, which are improved with exercise training (panel C). Created with BioRender.com.

**Figure 3 ijms-21-06948-f003:**
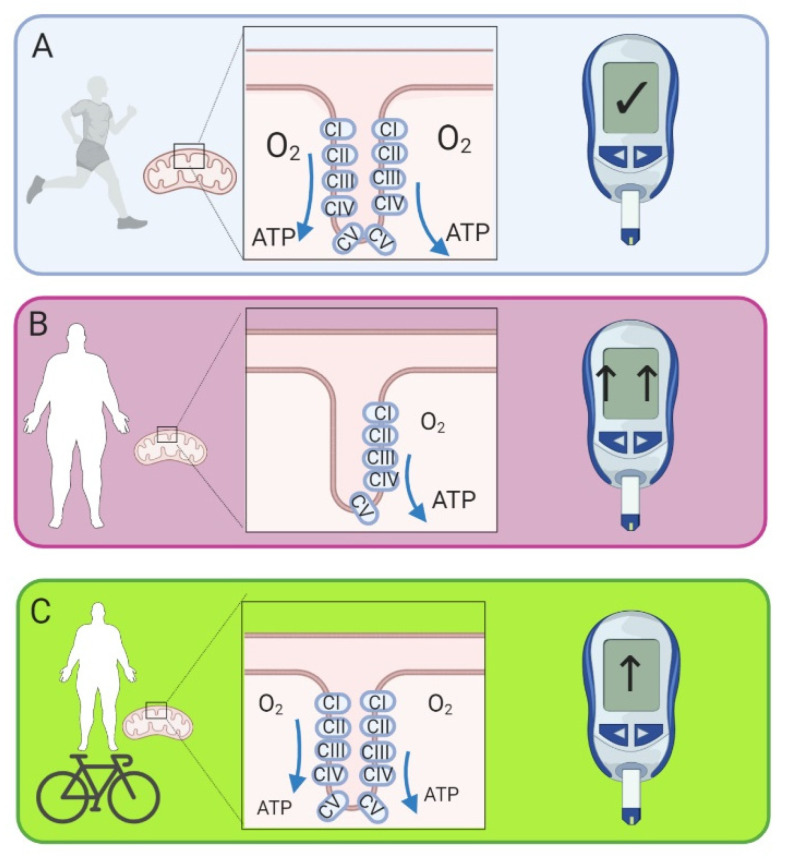
(**A**) Fit, healthy individuals have increased mitochondrial respiratory function and are insulin sensitive. (**B**) Patients with insulin resistance and type 2 diabetes (T2D) have comparatively lower mitochondrial respiratory function and are insulin resistant. (**C**) Patients with insulin resistance and T2D who are able to increase their physical activity see an improvement in mitochondrial respiratory function and a reduction in insulin resistance. Glucometer represents the degree of insulin sensitivity, and associated changes in blood glucose concentration. In panel (**A**), the glucometer shows blood glucose in the normal range, and fit, healthy people have normal insulin sensitivity. In panel (**B**) the glucometer shows the increased blood glucose concentrations present in people with insulin resistance, which are improved with exercise training (panel **C**). Created with BioRender.com.

**Table 1 ijms-21-06948-t001:** Mitochondrial content in patients with obesity and/or type 2 diabetes.

Study	Population	Method	Finding
Chomentowski et al., 2011 [36]	T2D Non-diabetic IR	TEM	↓ IMF, no change SS ↓ IMF, no change SS
Ritov et al., 2005 [7]	T2D	TEM	↓ SS
Kelley et al., 2002 [5]	Obese T2D	CS	No change ↓
Kim et al., 2000 [37]	Obese	CS	↓
Asmann et al., 2006 [32]	T2D	mtDNA	No change
Boushel et al., 2007 [38]	T2D	mtDNA	↑
Nair et al., 2008 [45]	T2D	CS	↓
van Tienen et al., 2012 [39]	T2D IR	CS	↓ No change
Mogensen et al., 2007 [43]	T2D	CS	No change
Holloway et al., 2007 [40]	Obese	CS	↓
Bruce et al., 2005 [44]	Obese	CS	No change

Abbreviations: CS: citrate synthase activity; IMF: intermyofibrillar; IR: insulin resistant; mtDNA: mitochondrial DNA; SS: subsarcolemmal; TEM: transmission electron microscopy; T2D: type 2 diabetes; ↓: lower, ↑: higher.

**Table 2 ijms-21-06948-t002:** Changes in mitochondrial content and function and insulin resistance with exercise training interventions in individuals who are insulin resistant, obese, and/or have T2D.

Study	Participants	Exercise Training	Insulin Resistance Outcome	Mitochondria Outcome
Short et al., 2003 [64]	Male and female, young and older participants (21–87 y) Healthy, low regular activity level, normal weight	16 weeks moderate-intensity exercise training	↑ insulin sensitivity only in younger participants	↑ mitochondrial gene expression ↑ mitochondrial enzyme activity
Menshikova et al., 2005 [52]	Male and female, overweight and obese, non-diabetic, sedentary	16 weeks, 60–70% maximal intensity for 30–40 min for 4–6 sessions per week	↑ insulin sensitivity	↑ activity of ETC enzymes
Bruce et al., 2006 [53]	Male and female, obese, sedentary non-diabetic	8 weeks, 5 days per week for 60 min at 65–70% of VO_2_ peak	↑ glucose tolerance	↑ fatty acid oxidation ↑ CPT1 activity ↑ β-HAD activity
Toledo et al., 2007 [48]	Sedentary, overweight/obese T2D	16–20 weeks moderate intensity	↑ insulin sensitivity	↑ mitochondrial content ↑ mitochondrial enzymes
Meex et al., 2010 [8]	Male T2D and healthy controls, overweight and obese, sedentary	12 weeks, 2 days per week For 30 min at 55% W_max_ aerobic exercise plus one session of resistance exercise per week—8 reps at 55% MVC and 2 series of 8 reps at 75% MVC	↑ insulin sensitivity	↑ mitochondrial function (^31^P-MRS)
Phielix et al., 2010 [9]	As for Meex et al.	As for Meex et al.	↑ insulin sensitivity	↑ mitochondrial function (HRR) ↑ mtDNA
Bordenave et al., 2008 [50]	Male T2D, overweight, sedentary	10 weeks, 2 days per week for 45 min at low-moderate intensity	No change in blood glucose	↑ lipid oxidation ↑ respiration ↑ CS activity
Little et al., 2011 [61]	T2D patients, obese, mostly sedentary	6 HIIT sessions over 2 weeks, 10 × 60 s intervals at 90% HR_max_	↓ hyperglycaemia ↑ GLUT4	↑ CS activity ↑ protein content of ETC complexes
Mogensen et al., 2009 [62]	Male T2D and controls, obese, similar activity levels in both groups (non-sedentary)	10 weeks, 5 days per week for 30 min moderate intensity interval and continuous training	↑ insulin sensitivity	↑ CS activity post exercise training, but not different in T2D to controls
Hey-Mogensen et al., 2010 [57]	Male T2D and controls, obese, non-sedentary	10 weeks, 4–5 days/week, moderate intensity	↑ insulin sensitivity	↑ respiration
Hood et al., 2011 [65]	Overweight, sedentary, non-diabetic	2 weeks, 3 days/week, HIIT	↑ HOMA, ↑ glucose transporter protein	↑ PGC-1α ↑ CS and COX-IV protein
Irving et al., 2011 [66]	Non-diabetic offspring of T2D parents and controls, sedentary	9 days intensive exercise training (continuous moderate and HIIT)	↑ insulin sensitivity in the controls only	↑ mitochondrial ATP production ↑ CS activity
Hutchison et al., 2012 [67]	Obese insulin-resistant women with PCOS and controls, sedentary	12 weeks, 3 days/week, moderate intensity and HIIT	↑ insulin sensitivity	No change in mitochondrial parameters
van Tienen et al., 2012 [39]	Obese control, pre-diabetic and T2D	1 year training in T2D participants (endurance and resistance)	ND	↑ ATP production ↑ Genes related to TCA cycle, β-oxidation, and oxidative phosphorylation
Coen et al., 2015 [68]	Men and women after RYGB surgery	Weight loss only or weight loss and 6 months exercise training (3-5 days/week, moderate intensity)	↑ glucose tolerance compared to weight loss only group	↑ respiration in exercise group
Konopka et al., 2015 [51]	Obese women with PCOS, and lean insulin-sensitive controls	12 weeks, 5 days per week, 60 min at 65% VO_2_ peak	↑ insulin sensitivity	↓ H_2_O_2_ emission
Axelrod et al., 2018 [58]	Obese, pre-diabetic, sedentary, male and female	12 weeks, 5 days per week, 60 min at 85% HR_max_	↑ insulin sensitivity	↑ PGC-1α ↑ CS activity
Kras et al., 2019 [69]	Obese and non-obese participants, sedentary, male and female	Single exercise session 45 min @ 65% HR reserve	↑ QUICKI ↓ plasma insulin in non-obese participants	↑ MAPR in IMF mitochondria, response less in SS mitochondria

Abbreviations: β-HAD: beta-hydroxyacyl CoA dehydrogenase, CPT1: carnitine palmitoyltransferase 1, CS: citrate synthase, COXIV: cytochrome c oxidase subunit 4, ETC: electron transport chain, HR: heart rate, HR_max_: heart rate maximum, HIIT: high intensity interval training, HOMA: homeostatic model assessment, HRR: high resolution respirometry, H_2_O_2_: hydrogen peroxide, IMF: intermyofibrillar, MAPR: mitochondrial ATP production rate, MVC: maximum voluntary contraction, ND: not determined, PCOS: polycystic ovary syndrome, PGC-1α: peroxisome proliferator-activated receptor gamma co-activator 1-alpha, ^31^P MRS: magnetic resonance spectroscopy, QUICKI: quantitative insulin sensitivity check index, RYGB: Roux-en-Y gastric bypass, SS: subsarcolemmal, TCA: tricarboxylic acid, T2D: type 2 diabetes, VO_2max_: maximal oxygen uptake, W_max_: maximal work load.

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
