# Peer review of "Are Alterations in Skeletal Muscle Mitochondria a Cause or Consequence of Insulin Resistance?"

_ijms, 2020, doi:10.3390/ijms21186948_

Round 1
Reviewer 1 Report
The review by Genders et al., “Are alterations in skeletal muscle mitochondria a cause or consequence of insulin resistance?”,ijms- 932105, is a very complete and updated presentation of the very broad and complex body of literature dealing with this very relevant and hot subject. The complexity of literature derives from the large amount of conflicting reports that have been summarized and exposed in a very clear and orderly manner. The aim of shedding some light on this difficult issue was pursued in a very elegant and convincing way. The only reason for not giving a clear answer to the title question was the real impossibility to drive a definite relationship between insuline resistance and alterations of skeletal muscle mitochondria. In fact, controversial results are presently too many and yet they have all been reported very precisely and thoroughly discussed. The use of English is great and sentences are very clear and easy to be read. The only change that has to be performed is in the format of the Ref. 81.
Author Response
Thank you to the reviewer for reading this manuscript and providing comments. We have fixed the formatting of reference 81.
Reviewer 2 Report
Dear Authors,
It is a very interesting review where authors highlighted a controversy regarding muscle mitochondria a cause or consequence of insulin resistance. Review is well written, and summed a significant available literature. I only have 2 minor concerns listed below:
- Graphical abstract is not clear. Authors showing that healthy individuals by exercise induces mitochondrial content, whereas inactivity leads to decline mito numbers. This does not represent the main content of this review. Please modify.
- Fig 2 and 3- A glucometer has been shown but not properly described in the text. please elaborate, and mention this part in figure as well as in text. Thanks
Author Response
Thank you to the reviewer for reading this manuscript and providing comments. We have corrected the graphical abstract (first page, line 23), and removed some references to mitochondrial content and physical activity/exercise training. We have also described the use of the glucometer in the figure legends of both figures 2 and 3 (lines 259-263 and 419-423).